# Whole Genome Sequencing and Phenotypic Analysis of Antibiotic Resistance in *Filifactor alocis* Isolates

**DOI:** 10.3390/antibiotics12061059

**Published:** 2023-06-15

**Authors:** Rosa Romero-Martínez, Anushiravan Maher, Gerard Àlvarez, Rui Figueiredo, Rubén León, Alexandre Arredondo

**Affiliations:** 1Department of Microbiology, DENTAID Research Center, 08290 Barcelona, Spain; 2Oral Surgery and Implantology, Faculty of Medicine and Health Sciences, University of Barcelona, 08036 Barcelona, Spain

**Keywords:** antimicrobial resistance genes, tetracycline, macrolides, *Filifactor alocis*, clinical isolates, subgingival, whole genome sequencing

## Abstract

There is scarce knowledge regarding the antimicrobial resistance profile of *F. alocis.* Therefore, the objective of this research was to assess antimicrobial resistance in recently obtained *F. alocis* clinical isolates and to identify the presence of antimicrobial resistance genes. Isolates were obtained from patients with periodontal or peri-implant diseases and confirmed by sequencing their 16S rRNA gene. Confirmed isolates had their genome sequenced by whole genome sequencing and their phenotypical resistance to nine antibiotics (amoxicillin clavulanate, amoxicillin, azithromycin, clindamycin, ciprofloxacin, doxycycline, minocycline, metronidazole, and tetracycline) tested by E-test strips. Antimicrobial resistance genes were detected in six of the eight isolates analyzed, of which five carried *tet*(32) and one *erm*(B). Overall, susceptibility to the nine antibiotics tested was high except for azithromycin in the isolate that carried *erm*(B). Moreover, susceptibility to tetracycline, doxycycline, and minocycline was lower in those isolates that carried *tet*(32). The genetic surroundings of the detected genes suggested their inclusion in mobile genetic elements that might be transferrable to other bacteria. These findings suggest that, despite showing high susceptibility to several antibiotics, *F. alocis* might obtain new antimicrobial resistance traits due to its acceptance of mobile genetic elements with antibiotic resistance genes in their genome.

## 1. Introduction

*Filifactor alocis* is an obligate anaerobic, non-sporeforming, Gram-positive rod [1,2]. This microorganism was first described in 1985 after being isolated from patients with periodontitis or gingivitis [3]. However, it was not until the early to mid-2000s, with the improvement of molecular identification techniques, when *F. alocis* was first acknowledged to be associated with oral diseases such as periodontitis or endodontic infections [4,5,6,7]. Later, the emergence of high-throughput sequencing allowed researchers to obtain a more realistic picture of the complexity of the oral microbiome and helped to confirm the strong association between *F. alocis* and periodontitis [8,9]. Moreover, *F. alocis* has been reported to employ several virulence strategies, including the invasion of epithelial cells, which induces the secretion of proinflammatory cytokines, subsequently triggering apoptosis [10,11]. The effects of *F. alocis* on the host cells also include the neutrophils, whose lifespan is extended through the inhibition of their apoptotic signaling while reducing their antimicrobial activities [2,12]. This elongates and worsens the inflammatory state of the periodontium, a situation in which *F. alocis* seems to thrive, due to the extended exposure of the periodontium to degradative enzymes secreted by neutrophils and the delayed initiation of tissue restorative mechanisms [12]. Other weapons in *F. alocis’* arsenal include (1) the production of exotoxins, such as FtxA, which belongs to the RTX family and might help damage the surrounding cells by forming pores in the membranes of epithelial cells [13], (2) the ability to overcome oxidative stress thanks to enzymes such as superoxide reductase, which helps *F. alocis* to overcome the neutrophil response [14], and (3) the capability of mechanically adapting to the environment by forming pili-like projections [15,16]. To make things worse, and similar to other periodontopathogens such as *Porphyromonas gingivalis*, *Fusobacterium nucleatum,* and *Aggregatibacter actinomycetemcomitans*, *F. alocis* has also been detected in non-oral environments such as lung and brain abscesses [17,18,19,20,21].

Therefore, it seems that *F. alocis* might have an important role in the occurrence of several infectious diseases, especially periodontitis. Nevertheless, little is known of its antimicrobial resistance (AMR) properties [22]. Given that most infections still need antimicrobials to resolve successfully [23], it seems crucial to have updated and thorough information regarding the levels of AMR as well as the genetic determinants that might confer such resistance. Therefore, the phenotypic resistance levels of *F. alocis* isolates, obtained from subgingival samples of patients with different oral diseases, were tested. Moreover, their genomes were screened for AMR genes and mobile genetic elements (MGEs) that might spread these genes.

## 2. Results

### 2.1. Identification of Isolates and Sequencing Analysis

A total of eight isolates of *F. alocis* were obtained from eight different patients, two with periodontitis, one with peri-implant mucositis, and five with peri-implantitis. Isolates were identified by their colony morphology and then confirmed using a combination of three sets of primers designed to amplify a specific region of the 16S rRNA gene of *F. alocis* and the *pilN* and the *ftxA* genes. All isolates were positive for the 16S rRNA and *pilN* amplicons, and all but the 14.12, 22.1, and 22.4 isolates were positive for *ftxA.* Sequencing of their genome yielded a total of 155,032,230 reads, adding up to 23,409,866,730 base pairs sequenced (mean per isolate: 19,379,028.8 reads and 2,926,233,341 base pairs; standard deviation: ±3,458,607.2 reads and ±522,249,686.6 base pairs).

### 2.2. Detection and Analysis of Antimicrobial Resistance Genes

Screening of the contigs revealed the presence of two ARM genes in six of the eight isolates (Table 1). The most prevalent AMR gene was *tet*(32) (5/8 isolates), which codes for a ribosomal protection protein (RPP) that confers resistance to tetracyclines, followed by *erm*(B) (1/8 isolates), a methyltransferase that confers resistance to macrolides. The genetic elements surrounding *tet*(32) were similar in all the isolates that carried this gene, suggesting the integration of *tet*(32) in an MGE (Figure 1). This MGE shares a high degree of homology with the MGE present in several bacterial species such as *Streptococcus anginosus* (accession number CP007573.1)*, Streptococcus oralis* (accession number CP097843.1), or *Clostridium scindens* (accession number CP036170.1), among others (data not shown). The contig containing *erm*(B) did not comprise the full sequence of the gene, which was cut at the 5′ ends, covering 72.2% of its aminoacidic sequence with an identity of 100% with a previously published sequence (accession number WP_001038795.1). Nevertheless, a transposase was detected just downstream *erm*(B) and homology of this contig with other sequences suggests that this gene was also integrated in an MGE that has been detected in strains of *S. anginosus* (accession number CP069892.1)*, Streptococcus salivarius* (accession number CP018189.1)*, Enterococcus saigonensis* (accession number AP022822.1), and *Enterococcus faecalis* (accession number CP118057.1)*,* among others (data not shown).

### 2.3. Antimicrobial Resistance Testing

Resistance to eight antibiotics pertaining to six different classes was tested using E-test strips. Susceptibility to these antibiotics was generally high (Table 2), with only one isolate (30.27) showing high levels of resistance, namely to azithromycin (AZM) (>256 µg/mL). Susceptibility or resistance to the antibiotics tested needs to be assessed with caution, since there are no established breakpoint concentrations for *F. alocis* in the European Committee on Antimicrobial Susceptibility Testing (EUCAST) guidelines [24]. Therefore, the MICs obtained must be analyzed by comparison with those of other anaerobic bacteria listed in the guidelines. According to this criterion, all isolates were susceptible to all the antibiotics tested except for isolate 30.27, which was resistant to clindamycin (CDM) and AZM.

## 3. Discussion

This study aimed to determine the antimicrobial resistance profiles of *F. alocis* isolates obtained from subgingival and peri-implant samples from patients with periodontal or peri-implant diseases. Their genome was sequenced to screen for AMR genes, and their phenotypical AMR was tested with the E-test method.

The significance of the role of *F. alocis* in periodontal and peri-implant diseases is increasingly being emphasized, as a growing number of studies utilizing molecular approaches are establishing its association with these diseases [25,26]. The fastidious conditions for growing this species have been a barrier to conduct studies based on its growth in isolation such as Whole Genome Sequencing or antimicrobial susceptibility analyses. In fact, only a study from 1985 [3] described the susceptibility of 20 *F. alocis* isolates to breakpoint concentrations of five antibiotics (chloramphenicol [12 µg/mL], clindamycin [1.6 µg/mL], erythromycin [3 µg/mL], tetracycline [6 µg/mL], and penicillin [2 U/mL]), observing that those isolates were susceptible to all breakpoint concentrations except for one isolate which was resistant to penicillin. Nevertheless, the aging of such data, the geographical constraints of the study, and the fact that the identity of those isolates remains to be confirmed with molecular approaches, call for new reliable data.

The results of this study show the high susceptibility of *F. alocis* to the most used antimicrobials in the clinical practice, with some exceptions such as isolate 30.27, which showed reduced susceptibility to CDM and extremely high tolerance to AZM; the ATCC 35896 strain, whose reduced susceptibility to tetracyclines has already been reported [22]. Resistance to tetracycline (TET) can be mediated through tetracycline resistance genes that code for RPPs, efflux pumps or inactivation enzymes [27]. Tolerance to this class of antibiotics by the ATCC 35896 strain has been linked to the presence of the *tet*(M) gene, which codes for an RPP and is enclosed in an MGE [28]. However, none of the isolates in this study carried *tet*(M). Instead, five of them carried *tet*(32) (Table 1), which was integrated into another MGE that shared a highly homologous 12-kb region with strains of *S. anginosus, S. oralis,* and *C. scindens*, among others, which also contain *tet*(32) in their genome. Moreover, the presence of this gene in the isolates seemed to increase their tolerance to minocycline (MIN), doxycycline (DOX), and even more to TET when comparing with the isolates that did not carry it (5.15, 48B, and 30.27), despite not reaching the levels conferred by *tet*(M) in the ATCC 35896 strain (Table 2). *tet*(32) codes for an RPP that binds to a ribosome that is being altered by a molecule of TET, or one of its derivatives, and breaks the union to the detriment of a GTP molecule [27]. Both TetM and Tet32 use this same mechanism, which suggests that the different levels of resistance of the isolates that carry one gene or the other may be due to something else, such as for example, different levels of gene expression or the impact that genetic surroundings might have on these genes. In a recent study, the genes *tet*(M) and *tet*(32) were the two most detected TET resistance genes among subgingival isolates obtained from patients with periodontitis [29]. However, to our knowledge, this is the first report of *tet*(32) in bacteria from the genus *Filifactor.* The high prevalence in the oral environment of these genes, which are often embedded in MGEs, such as conjugative transposons [30,31], might be an indicator of their ease of spread, which does not seem to bend under phylogenetic barriers and could render the use of tetracyclines useless in the oral environment.

*erm*(B) codes for a methyltransferase that methylates an adenine in the domain where macrolides and lincosamides bind to the 23S rRNA region of the major subunit of the bacterial ribosome, preventing such binding and thus conferring resistance [32,33]. Therefore, the presence of this gene might have provided isolate 30.27 with a very high tolerance to AZM (>256 µg/mL) and reduced its susceptibility to CDM (1 µg/mL). The present study is, to our knowledge, the first report of a macrolide resistance gene in *F. alocis*. As observed in TET resistance genes, *erm*(B) shows a high prevalence in oral bacteria and can be detected in conjugative transposons [29,34]. In fact, in a study from 2019, the authors also detected *erm*(B) for the first time in bacteria from the genus *Prevotella* [35], which until that moment were only known to carry *erm*(A), *erm*(C), *erm*(F) and *erm*(G) [36], highlighting the transferrable properties of these genes through MGEs that carry them. Interestingly, a previous study pointed out the high resemblance of a 45-kb region of the *Streptococcus dysgalactiae* subsp. *equisimilis* containing *erm*(B), with the region of *F. alocis* ATCC 35,896 that contains *tet*(M), suggesting that a swap between *tet*(M) and *erm*(B) took place at some time [28]. However, the surroundings of *erm*(B) in isolate 30.27 showed little resemblance with either *F. alocis* ATCC 35896 or the strain of *S. dysgalactiae* of the mentioned study. Given its close proximity to the *tnpA* and *int* transposases, also present in conjugative transposons such as Tn3, Tn916, Tn1721, Tn1545 and Tn6261 among many others [37], and the high homology of the 9-kb downstream of the gene *erm*(B) with a region of *S. anginosus*, *S. salivarius, E. saigonensis* and, to a lesser extent, *E. faecalis*, it is highly probable that this gene is being carried within an MGE.

This study is not exempt from limitations. For instance, due to the fastidious culture, identification, and isolation involved, the number of isolates was not high enough to extract conclusions of AMR in *F. alocis*, even more given the variability observed in the eight isolates of the study and despite working with a similar or higher number of isolates than other publications [13,22]. Moreover, full assemblies of the genomes using long-read amplicon sequencing in conjunction with shotgun sequencing might help to better understand those regions that were truncated due to methodological limitations. On the other hand, the reduced sample size did not allow to make comparisons between the isolates found in the different pathologies (periodontitis, peri-implant mucositis, or peri-implantitis). Further studies are needed to assess the contribution of the AMR genes detected to the phenotypic resistance observed in the study isolates.

AMR in bacteria that have a potential role in infectious diseases should be strictly monitored [38]. Although its role in periodontal disease is still largely unknown, evidence that *F. alocis* is strongly associated with periodontitis is growing [17,25,26]. This study shows that *F. alocis* isolated from either subgingival or peri-implant samples is still highly susceptible to the most common antibiotics. However, AMR genes embedded in MGEs were observed in almost all isolates, and one of them showed a very high tolerance to AZM, which highlights the need to keep a close eye on the AMR profiles in these microorganisms if antibiotics are to be kept as a treatment option.

## 4. Material & Methods

### 4.1. Sample Collection and Culturing Conditions

Patients diagnosed with periodontitis, peri-implant mucositis or peri-implantitis attending the Dental Hospital of the University of Barcelona, were asked to volunteer for 2 studies that aimed at determining the presence of *Filifactor alocis* in periodontitis and peri-implant diseases. The study protocols were approved by the local Ethics Committee (protocols ref. 15/2022 and 36/2022). All volunteers signed an Institutional review board-approved informed consent form. Subgingival and peri-implant samples were obtained by placing sterile paper points for 30 s in subgingival pockets. Then, paper points were pooled in a vial containing either 1.5 mL of cold sterilized reduced transport medium (RTF) without ethylenediaminetetraacetic acid or modified Amies medium containing 20.9 g/L Amies transport with charcoal (Condalab^TM^, Madrid, Spain), cysteine (0.012 g/L), tryptose (0.5 g/L) and peptone (0.5 g/L). Both mediums were incubated for 24 h in anaerobic conditions (10% H_2_, 10% CO_2_ and 80% N_2_) at 37 °C prior to the sample collection. Samples were sent to the Dentaid Research Center (Cerdanyola del Vallès, Spain) to be processed.

Samples in Amies medium were resuspended in 400 µL of phosphate-buffered saline (PBS), transferred to a new 1.5 mL sterile centrifuge tube, and vortexed for 1 min. Samples in RTF were directly vortexed and 400 µL were transferred to a new 1.5 mL sterile centrifuge tube. Then, serial dilutions were made and plated on blood agar plates containing 40 g/L Oxoid Nutrient Broth No. 2, 5% horse blood, hemin (5 mg/L) and menadione (1 mg/L) and were incubated under anaerobic conditions at 37 °C for 10 days.

Colonies suspected of being *F. alocis* were isolated, replated in blood agar plates, and incubated in anaerobic conditions for 7 days. Then, they were recovered under a stereo microscope to ensure correct identification. Translucent flattened colonies with a wide halo and small round-shaped center were considered to be *F. alocis*, isolated on new plates and cryo-conserved at −80 °C. In order to extract their DNA and to perform the antimicrobial susceptibility tests, 5–10 colonies were inoculated in liquid cultures with BHI-GA medium containing 37 g/L Brain Heart Infusion (Becton Dickinson), yeast extract (1 g/L), sodium bicarbonate (2 g/L), L-cysteine (1 g/L), L-arginine (2 g/L), and noble agar (1 g/L). Liquid cultures were grown under anaerobic conditions at 37 °C for at least 3 days.

### 4.2. DNA Isolation and Polymerase Chain Reaction-Based Characterization

DNA extractions were performed using the QIAamp DNA Mini Kit (Qiagen, Hildem Germany) according to the manufacturer’s instructions. DNA was quantified using a NanoDrop 2000C UV-vis Spectrophotometer (NanoDrop Technologies, Wilmington, DE, USA). PCR reactions were made with 3 different pairs of oligonucleotides. First, DNA amplicons of the 16S rRNA gene of *F. alocis* were obtained as described [5]. Then, a second reaction was performed for *pilN*, a specific gene of *F. alocis*, with forward primer (*pilN*_F: 5′-GCTCAGCAAACATGCGATTG-3′) and reverse primer (*pilN*_R: 5′-GAAGGCTATGATTTGATTGTTTCC-3′) to amplify a 156-bp-length fragment. Finally, a third reaction was conducted for the *ftxA* gene to amplify a 798-bp-length fragment as described [39]. A total of 0.5 µM was the final concentration used for all primers. Between 30 and 100 nanograms of DNA were used, together with 1× PCR buffer, 1× dNTPs solution, 2.5 mM of MgCl_2,_ and 1 unit of Taq polymerase (Takara Taq^TM^ DNA Polymerase). PCR cycling conditions were 10 min at 94 °C, followed by 30 cycles of 94 °C for 30 s, 60 °C for 30 s, and 72 °C for 40 s, followed by a final extension of 10 min at 72 °C, performed in a T3000 Thermocycler (Biometra, Göttingen, Germany). PCR products were assessed by electrophoresis in 2% agarose gels with SybrSafe^TM^ DNA Gel Stain (Life Technologies, Carisbad, CA, USA).

### 4.3. Antibiotic Susceptibility Testing

To determine antibiotic susceptibility, strain ATCC 35896 and the *F. alocis* clinical isolates were plated on blood agar, E-test strips (BioMérieux^TM^, Marcy-l’Étoile, France) were added and then plates were incubated at 37 °C in anaerobic conditions. E-test strips contained a concentration gradient of amoxicillin, amoxicillin with clavulanic acid, MIN, DOX, TET, metronidazole, AZM, CDM, or ciprofloxacin. The minimum inhibitory concentration of all the antibiotics was measured as instructed by the manufacturer. *Streptococcus pneumoniae* ATCC 49619 and *Escherichia coli* DSM 1576 were used as quality control strains.

### 4.4. Whole-Genome Sequencing

DNA from the isolates (5.15, 8.25, 11.40, 14.12, 22.10, 22.40, 30.27, and 48B) was sent to Macrogen, Inc. (Seoul, South Korea) for Whole-Genome Sequencing using the Nextera DNA XT Library and the NovaSeq6000 platform (Illumina, San Diego, CA, USA). Quality of the reads was assessed using FastQC (https://www.bioinformatics.babraham.ac.uk/projects/fastqc/ (accessed on 1 March 2023)) and trimmed using Cutadapt [40]. Trimmed reads were assembled using Bowtie2 and using the *F. alocis* ATCC 35896 assembly available at Genbank (accession number GCA_000163895.2) as a reference genome. Genomes were screened for AMR genes using AMRFinderPlus [41] fed with contigs assembled with SPAdes 3.15.4 [42]. DNA regions of interest were further analyzed using de NCBI ORF finder (https://www.ncbi.nlm.nih.gov/orffinder/ (accessed on 5 March 2023)) and BLAST (https://blast.ncbi.nlm.nih.gov/Blast.cgi (accessed on 5 March 2023)).

## Figures and Tables

**Figure 1 antibiotics-12-01059-f001:**
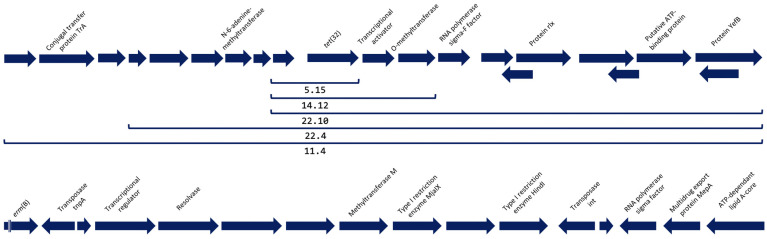
Open Reading Frames of the surroundings of the antimicrobial resistance genes detected. At the top of the figure, the genetic surroundings of *tet*(32) and the length of the construction that was detected in each of the isolates indicated with a bracket. At the bottom of the figure, the genetic surroundings downstream gene *erm*(B). This gene was truncated in the contig and approximately 30% of its upstream sequence is missing, as well as its surroundings.

**Table 1 antibiotics-12-01059-t001:** Antimicrobial resistance genes detected in the isolates. Reference sequences of *tet*(32) and *erm*(B) can be accessed with the accession numbers WP_003505402.1 and WP_001038795.1, respectively.

Isolate	Gene	Class of Antimicrobial	Coverage of Sequence (%)	Identity of Sequence (%)
5.15	*tet*(32)	Tetracycline	68.39	99.54
11.40	*tet*(32)	Tetracycline	100	100
14.12	*tet*(32)	Tetracycline	100	100
22.10	*tet*(32)	Tetracycline	100	100
22.40	*tet*(32)	Tetracycline	100	100
30.27	*erm*(B)	Macrolide	72.24	100

**Table 2 antibiotics-12-01059-t002:** Minimum inhibitory concentrations (µg/mL) of the tested *F. alocis* isolates. Antibiotic concentrations of the E-test strips ranged from 0.016 to 256 µg/mL in all antibiotics with the exception of ciprofloxacin, which ranged from 0.002 to 32 µg/mL. AMC: amoxicillin clavulanate, AMX: amoxicillin, AZM: azithromycin, CDM: clindamycin, CIP: ciprofloxacin, DOX: doxycycline, MIN: minocycline, MTZ: metronidazole, TET: tetracycline.

Isolate	AMC	AMX	AZM	CDM	CIP	DOX	MIN	MTZ	TET
5.15	<0.016	<0.016	1	0.047	0.008	0.047	0.032	<0.016	0.25
8.25	0.016	0.023	0.032	0.016	0.023	0.023	0.016	0.016	0.016
11.40	0.032	0.023	0.047	<0.016	0.032	0.125	0.094	<0.016	0.75
14.12	0.25	<0.016	0.032	<0.016	0.012	0.5	0.125	<0.016	0.75
22.10	0.016	0.064	0.047	0.016	0.023	0.25	0.19	0.016	1
22.40	0.023	0.032	0.032	<0.016	0.016	0.25	0.032	<0.016	0.75
30.27	<0.016	<0.016	>256	1	0.125	<0.016	< 0.016	<0.016	0.016
48B	0.032	0.016	0.064	<0.016	0.047	0.047	< 0.016	0.016	0.016
ATCC 35896	0.064	0.032	0.032	<0.016	0.023	1	1	0.016	2
Range	<0.016–0.094	<0.016–0.25	0.032–>256	<0.016–0.19	0.08–1	<0.016–0.75	<0.016–0.19	<0.016–>256	0.016–1
MIC_50_	0.016	0.023	0.047	<0.016	0.023	0.047	0.032	<0.016	0.25
MIC_90_	0.032	0.064	1	0.047	0.047	0.25	0.125	0.016	0.75

## Data Availability

Data might be shared upon reasonable request.

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
