# Peer review of "Whole Genome Sequencing and Phenotypic Analysis of Antibiotic Resistance in Filifactor alocis Isolates"

_antibiotics, 2023, doi:10.3390/antibiotics12061059_

Round 1

Reviewer 1 Report

This study was done to assess the antimicrobial resistance profile of F. alocis obtained from clinical isolates (patients with periodontal or peri implant diseases) and to identify the presence of antimicrobial resistance genes. Whole genome sequencing and their phenotypical resistance to 9 antibiotics (amoxicillin clavulanate, amoxicillin, azithromycin, clindamycin, ciprofloxacin, doxycycline, minocycline, metronidazole, and tetracycline) were tested by E-test strips. 

The manuscript on this particular topic is well-written and the subject chosen is fairly original. The authors did a good job of describing the methodology and the results seem well justified. 

Author Response

We would like to thank the reviewer for the effort reviewing the manuscript and for the kind words. 

Reviewer 2 Report

This is a good study looking at an uncommon bacterium F. alocis, it will definitely contribute to knowledge especially in AMR-related to oral pathology.

The only recommendation I have is that you can explore phylogenetic relationships with other related isolates in the public domain (if possible). This may provide more insight into the biology of this bacterium.

Author Response

We would like to thank the reviewer for the effort reviewing the manuscript. Regarding the recommendation proposed by the reviewer, we reckon that it would be very interesting to be able to associate phylogenetic relationships of the isolates that we have recovered with AMR data and with other F. alocis isolates. However, it has to be noted that the number of sequenced genomes of F. alocis is still low. There is only one fully assembled genome available (accession number: GCA_000163895.2) and two genomes at contig level (accession numbers: GCA_905372415.1 and GCA_937959765.1) which hinders our ability to perform such study. Moreover, we did not have enough coverage to assemble the genomes, keeping our data at the contig/scaffold level. It is true that phylogenetic relationships can be studied based on particular genes that might not need a full assembly of the genome. Nevertheless, we are planning to extend our knowledge of F. alocis at several levels, which will include the resequencing of these isolates, and many more, using long read sequencing techniques. This will allow us to perform full assemblies of the sequenced genomes, at which point we will certainly perform phylogenetic analysis between the isolates.

Reviewer 3 Report

I have some question and suggestions for the authors

1. Is authors submitted WGS data to gene bank? If yes then please provide accession number.

2. MIC data in table 2 is not well described. Please provide original data of MIC and also mention drug concentration range in text.

3. Further bioinformatic analysis can be performed from WGS

Author Response

I have some question and suggestions for the authors

First, we would like to thank the reviewer for the effort reviewing this manuscript. We will try our best to address the reviewer’s concerns.

  1. Is authors submitted WGS data to gene bank? If yes then please provide accession number.

We have not submitted our sequencing results to GenBank and therefore we cannot provide the accession numbers. We are planning to expand our knowledge on F. alocis and therefore we are going to resequence these isolates, together with others that we are still currently isolating, with long read sequencing to obtain assembled genomes. Since we truly believe in sharing all the data from sequencing data, we will make it available to the scientific community as soon as we have it ready. Given that we should be able to upload fully assembled genomes in the near future, we chose not to upload incomplete data in order to avoid adding information that in the end could become noise in the databases. Nevertheless, if the reviewer considers that we should upload this data, we are open to do it.

  1. MIC data in table 2 is not well described. Please provide original data of MIC and also mention drug concentration range in text.

We are not sure to understand the reviewer’s concern. We have followed the format that can be found in other studies published in the same journal (for example: https://www.mdpi.com/2079-6382/12/6/993 or https://www.mdpi.com/2079-6382/12/6/953) and other journals (doi: 10.1111/odi.13043, doi: 10.1093/jac/dkad168 or doi: 10.3389/fcimb.2023.1141115). Nevertheless, we have added the range of the E-test strips for each antibiotic in the table header (highlighted in red so it is easier to review), which is what we understand the reviewer is requesting. If the reviewer considers that more information is needed, please let us know so we can add it.

  1. Further bioinformatic analysis can be performed from WGS

We agree with the reviewer that the data obtained through WGS can be used to obtain a lot of information from the isolates through bioinformatic tools. However, given the scope of the special issue (Periodontitis: Prevention and Treatment) and the context and objectives of the manuscript, we fail to identify other bioinformatic analysis that might add to the current manuscript. Nevertheless, and as previously commented, we are planning to conduct further analysis that will expand our knowledge of F. alocis. In such study, we expect that the information obtained through bioinformatic analysis of WGS will be very valuable to explain the traits of this microorganism.